# Unfavorable Reduction in the Ratio of Endothelin B to A Receptors in Experimental 5/6 Nephrectomy and Adenine Models of Chronic Renal Insufficiency

**DOI:** 10.3390/ijms21030936

**Published:** 2020-01-31

**Authors:** Suvi Törmänen, Päivi Lakkisto, Arttu Eräranta, Peeter Kööbi, Ilkka Tikkanen, Onni Niemelä, Jukka Mustonen, Ilkka Pörsti

**Affiliations:** 1Faculty of Medicine and Health Technology, Tampere University, 33520 Tampere, Finland; 2Minerva Institute for Medical Research, 00290 Helsinki, Finland; 3Clinical Chemistry and Hematology, University of Helsinki and Helsinki University Hospital, 00014 Helsinki, Finland; 4Eye Centre, Tampere University Hospital, 33520 Tampere, Finland; 5Abdominal Center, Nephrology, University of Helsinki and Helsinki University Hospital, 00014 Helsinki, Finland; 6Department of Clinical Chemistry and Medical Research Unit, Seinäjoki Central Hospital, 60220 Seinäjoki, Finland; 7Department of Internal Medicine, Tampere University Hospital, 33520 Tampere, Finland

**Keywords:** chronic renal insufficiency, chronic kidney disease, endothelin receptor A, endothelin receptor B, creatinine, calcium, phosphate, uric acid, sitaxentan, cinacalcet, paricalcitol

## Abstract

Chronic renal insufficiency (CRI) is characterized by increased endothelin 1 (ET-1) synthesis. We studied rat kidney endothelin receptor A (ETA) and receptor B (ETB) expressions after 12 and 27 weeks of 5/6 nephrectomy, and after 12 weeks of 0.3% adenine diet, representing proteinuric and interstitial inflammation models of CRI, respectively. Uric acid and calcium-phosphate metabolism were modulated after 5/6 nephrectomy, while ETA blocker and calcimimetic were given with adenine. Endothelin receptor mRNA levels were measured using RT-qPCR and protein levels using autoradiography (5/6 nephrectomy) or ELISA (adenine model). Both 12 and 27 weeks after 5/6 nephrectomy, kidney cortex ETA protein was increased by ~60% without changes in ETB protein, and the ETB:ETA ratio was reduced. However, the ETB:ETA mRNA ratio did not change. In the adenine model, kidney ETA protein was reduced by ~70%, while ETB protein was suppressed by ~95%, and the ETB:ETA ratio was reduced by ~85%, both at the protein and mRNA levels. The additional interventions did not influence the observed reductions in the ETB:ETA ratio. To conclude, unfavorable reduction in the ETB:ETA protein ratio was observed in two different models of CRI. Therefore, ETA blockade may be beneficial in a range of diseases that cause impaired kidney function.

## 1. Introduction

Chronic kidney disease (CKD) is a global challenge and presents an urgent demand for novel strategies to hinder its progression [1]. Promising experimental and clinical results suggest that, even on top of renin-angiotensin system (RAS) inhibition, selective endothelin receptor A (ETA) antagonists may improve the prognosis of CKD [2,3,4]. A recent review provided a comprehensive overview of the potential renoprotective effects of ETA antagonists [5].

Endothelin 1 (ET-1) has a pivotal role in renal physiology as a modulator of glomerular filtration rate (GFR) as well as solute and water balance [6]. However, pathological endothelin system (ET-system) activation promotes cell proliferation, inflammation and extracellular matrix accumulation. ET-1 upregulation also increases intraglomerular pressure and glomerular permeability [2,6,7]. Regardless of the initial renal insult, systemic and renal ET-1 production is increased in various forms of CKD, and the production inversely correlates with the level of GFR [2,8]. The causality between pathological ET-system activation and renal damage is supported by models where transgenic rodents expressing human ET-1 or endothelin 2 (ET-2) genes predominately in the kidney develop glomerulosclerosis, tubulointerstitial fibrosis, and renal insufficiency in the absence of systemic hypertension [9,10].

ET-1 acts mainly in an autocrine or paracrine manner via two G-protein coupled receptors, endothelin receptor A (ETA) and endothelin receptor B (ETB). These receptors mostly induce opposing effects, whereby both ET-1 up-regulation and the ratio between ETB and ETA expression determine the biological effects of the ET-system [2,8]. Briefly, ETA promotes vasoconstriction, inflammation and fibrogenesis, while ETB mediates vasodilatation, diuresis and natriuresis. Therefore, ETB counteracts the harmful effects of ETA, and serves as a scavenger receptor for active ET-1 [2,8]. Under physiological conditions, 70% of the ET-receptors in both the cortex and medulla of the human kidney are ETB, whereas ETA is the predominant form in the smooth muscle of the renal vasculature [11].

In addition to ET-1 upregulation, CKD is associated with putative changes in the ETB versus ETA balance [2]. Elevated ET-1 concentration in CKD may result from both increased production and decreased elimination, as reduced ETB scavenger receptor activity might explain the increased ET-1 concentration in CKD [12]. Furthermore, ETB receptor activation increases endothelial nitric oxide (NO) release, which counteracts the vasoconstrictive effects of ET-1 and also down-regulates ET-1 production [13]. Vice versa, ET-1 acting via ETA has been linked to the development of CKD-associated endothelial dysfunction with diminished bioavailability of NO [14,15]. The strong counter-play between ETA and NO in CKD pathophysiology was recently supported by a rat ischemia-reperfusion injury study, where ETA down-regulation with small interfering RNA (siRNA) treatment resulted in reduced creatinine levels and inflammatory factors and alleviated renal histological damage, whereas NO expression was increased and ET-1 expression reduced. Interestingly, the renoprotective effects of ETA down-regulation were reversed by treatment with a NO-synthase inhibitor [16].

The shift in the ET-receptor balance was indirectly supported by the experimental and clinical studies which showed that selective ETA blockade is usually more advantageous in CKD, and that concomitant ETB receptor blockade might even mitigate the renoprotective effects of ETA antagonists [8,11]. ETA antagonists ameliorate renal injury, proteinuria, and disease progression in experimental diabetic, hypertensive, and remnant kidney rat models of chronic renal insufficiency (CRI) [2,8]. Recently, we reported that the highly selective ETA antagonist sitaxentan ameliorated the progression of experimental adenine-induced CRI by protecting renal histology and reducing the decline in GFR. Importantly, the adenine model did not induce significant glomerular changes or proteinuria, indicating the potential therapeutic value of ETA blockade beyond itsantiproteinuricactions. This suggests that ETA blockade may also be beneficial in tubulointerstitial renal diseases [17].

The pathophysiological changes in ETB and ETA receptor expression in CKD remain to be established due to contradictory findings in previous studies [18,19,20,21,22,23,24,25,26,27,28,29]. Reports about the expression of the ETB and ETA proteins are particularly scarce. Here we focused on ETB and ETA mRNA and protein expressions, and the ETB:ETA ratio, in the 5/6 nephrectomy model (12 and 27 weeks of follow-up) and the adenine-induced model (12 weeks of follow-up), representing proteinuric and tubulointerstitial inflammatory models of CRI, respectively. We tested the hypothesis whether there are changes in the ETB or ETA expressions or the ETB:ETA ratio in these experimental models of kidney damage, which would suggest that these mechanisms contribute to the pathophysiology of CRI irrespective of the initial cause of renal damage. Additionally, uric acid and calcium-phosphate metabolism were modulated in the 5/6 nephrectomy model, while ETA blocker and calcimimetic were given in the adenine model.

## 2. Results

### 2.1. Twelve-Weeks of Chronic Renal Insufficiency Induced by 5/6 Nephrectomy

Three weeks after sham or 5/6 nephrectomy operation (NX), rats received either the control diet or the 2.0% oxonic acid diet for 9 weeks. The groups and the animal numbers were Sham (*n*=12), Sham+Oxo (*n*=12), NX (12) and NX+Oxo (*n*=12). We previously published the study protocol in more detail, and essential demographic and laboratory results of the study groups were included in the previous reports [30,31,32].

The rats receiving oxonic acid had a lower body weight at the end of the study (Table 1). Blood pressure (BP) was elevated by 5/6 nephrectomy when compared with the sham-operated groups. Creatinine clearance was reduced in both NX groups and, as expected, plasma uric acid was increased in Sham+Oxo and NX+Oxo rats when compared with the corresponding groups not receiving oxonic acid. There were no differences in plasma calcium concentrations between the groups, while plasma phosphate values were increased by 5/6 nephrectomy (Table 1).

Urine protein excretion was increased by 5/6 nephrectomy and decreased by oxonic acid treatment (Table 1). Kidney tubulointerstitial damage index was increased in the NX and NX+Oxo groups when compared with the Sham and Sham+Oxo groups, respectively. However, the NX+Oxo group had a lower tubulointerstital damage index than the NX group. ETB mRNA expression was reduced by both 5/6 nephrectomy and oxonic acid treatment. ETA mRNA was reduced by 5/6 nephrectomy, but no significant differences were detected in the ratio of ETB to ETA mRNA. Also, no interactions between the 5/6 nephrectomy and oxonic acid treatment were detected in the analyses of any of the variables (Table 1).

In the renal medulla, ETB protein expression remained unchanged, while ETA was 1.4-fold higher in both the NX and NX+Oxo groups when compared with the Sham and Sham+Oxo groups, respectively (Figure 1A,B). The 5/6 nephrectomy reduced the ratio of the medullary sham-related ETB:ETA protein expression (Figure 1C).

In the renal cortex, ETB protein expression was not changed either by 5/6 nephrectomy, while ETA protein expression was approximately 1.6-fold higher in the NX and NX+Oxo groups than in the Sham and Sham+Oxo groups, respectively (Figure 1D,E). Cortical ETB protein expression was slightly increased by oxonic acid treatment (Figure 1D). The ratio of the cortical sham-related ETB:ETA protein expression was reduced in both NX groups (Figure 1F). The representative original tracings of ETB and ETA autoradiography from the sham-operated and NX rats are depicted in Figure 2.

### 2.2. Twenty-Seven Weeks of Chronic Renal Insufficiency Induced by 5/6 Nephrectomy

Fifteen weeks after sham or NX operation, the NX rats were divided into four groups receiving diets containing either 0.3% Ca + 0.5% Pi, 3.0% Ca + 0.5% Pi, 0.3% Ca + 1.5% Pi or 0.3% Ca + 0.5% Pi combined with intraperitoneal paricalcitol injections (100 ng/rat three times a week). The final animal numbers in the study groups were Sham (*n* = 13), NX (*n* = 7), NX+Ca (*n* = 11), NX+Pi (*n* = 7) and NX+Pari (*n* = 9), respectively. The study protocols and the effects of calcium-phosphate balance modulation on RAS components have been previously published in more detail, and the main laboratory results of the study groups were also included in the previous reports [33,34,35,36].

The body weights of the NX and Sham groups did not differ at the end of the study, and there were no significant differences in the body weights between the additional intervention groups and the NX group (Table 2). BP was significantly elevated in the NX group when compared with the Sham group, while the NX+Ca group had a lower BP than the NX group. Creatinine clearance was reduced, and creatinine and urea were increased in the NX groups when compared with the Sham groups, and there were no statistically significant differences between the NX group and the additional intervention groups in variables reflecting renal function (Table 2). 

The NX group had reduced 25OH-D_3_ and 1,25(OH)_2_D_3_ when compared with the Sham group (Table 2). NX+Pari rats had further suppressed 1,25(OH)_2_D_3_ levels when compared with the NX group. Ionized calcium was increased in NX+Ca rats and reduced in NX+Pi rats when compared with the NX group. The NX group had higher phosphate than the Sham group, whereas the NX+Ca group had lower plasma phosphate concentration than the NX group. Parathyroid hormone (PTH) was significantly elevated in the NX group when compared with the Sham group, while NX+Ca rats had suppressed PTH, and NX+Pi rats had further increased PTH when compared with the NX group (Table 2).

Urine protein excretion was higher in the NX group than in the Sham group, and this further increased in NX+Pi rats when compared with the NX group (Table 2). As expected, the tubulointerstitial damage index was increased in the NX group when compared with the Sham group. However, the tubulointerstitial damage index was lower in the NX+Ca rats than in the NX group. 

ETB and ETA mRNA were both increased in the NX group when compared with the Sham group. The ratio between ETB and ETA mRNA expressions did not differ between any of the groups (Table 2).

In the renal medulla, the sham-related ETB protein expression did not differ between any of the groups (Figure 3A). The medullary ETA protein expression was numerically ~1.5-fold higher in the NX group than in the Sham group, but the difference was not significant (*p* = 0.088) (Figure 3B). The sham-related medullary ETB:ETA protein expression was also numerically lower in the NX group than in the Sham group, but this was not significant, either (*p* = 0.066) (Figure 3C). 

In the renal cortex, the sham-related ETB protein expression did not differ between the groups, whereas ETA protein expression was ~1.6-fold higher in the NX group when compared with the Sham group (Figure 3D,E). The sham-related ETB:ETA protein expression ratio in the renal cortex was significantly reduced in the NX group when compared with the Sham group and there were no differences between the additional intervention groups and the NX group (Figure 3F).

### 2.3. Twelve Weeks of Chronic Renal Insufficiency Induced by 0.3% Adenine Diet

The study groups and the final animal numbers were the control (Control, *n* = 12), adenine (Ade, *n*=20), adenine+sitaxentan (Ade+S, *n* = 16), adenine+cinacalcet (Ade+C, *n* = 16), and adenine+sitaxentan+cinacalcet treatments (Ade+SC, *n* = 16). We have previously reported the study protocol in more detail, the kidney morphology, and the principal laboratory results of the study [17]. The body weight of the Ade group was reduced at the end of the study when compared with the Control group (Table 3). The adenine group showed no elevation of BP by the end of the study. The Ade+S group had lower BP than the Ade group, and Ade+SC rats had significantly lower BP than Ade+C rats. The Ade group had markedly reduced creatinine clearance when compared with the Control group, while the reduction was ameliorated in the Ade+S group when compared with the Ade group. 

Plasma creatinine and urea were increased in the Ade group when compared with the Control group (Table 3). The increase was significantly lower in the Ade+S group when compared with the Ade group. The plasma calcium concentration was not changed in any other groups, except for the fact that the Ade+SC rats had slightly lower calcium level than Ade+S rats. Phosphate levels were increased in the Ade group when compared with the Control group, while Ade+S rats had slightly lower phosphate level than Ade+SC rats. Plasma PTH was increased in the Ade group when compared with the Control group. PTH was reduced in the Ade+S group and especially in the Ade+C group compared with the Ade group. Ade+SC rats had a lower PTH when compared with the Ade+S group. Urine protein excretion remained unchanged in the Ade group when compared with the Control group and the additional interventions had no effects on this variable (Table 3).

ETA mRNA was increased, while ETB mRNA and the ETB:ETA mRNA ratio were reduced in the Ade group when compared with the Sham group (Table 3). The ETB and ETA protein levels were suppressed by ~95% and ~70%, respectively, in the Ade group when compared with the Control group (Figure 4A,B). In the Ade group, the ETB:ETA expression ratio was reduced by ~85% both at the mRNA and protein levels (Table 3 and Figure 4C). No significant differences were detected in the post-hoc comparisons of ET-receptor expressions between the Ade+S, Ade+C and Ade groups, or between the Ade+SC and the Ade+S or Ade+C groups.

Comparisons between the adenine-treated groups: additional analyses using two-way analysis of variance(ANOVA): When only the adenine-receiving groups were analyzed using two-way ANOVA, the outcome of the results presented in Table 3 were corresponding to the primary analyses described above. Only the ETB vs. ETA ratio showed a minor increasing effect by cinacalcet. In ET-receptor protein expressions, some effects were detected: thecombined sitaxentan+cinacalcet slightly increased ETB protein (*p* = 0.010), while the effects of sitaxentan and cinacalcet alone on ETB protein were of borderline significance (*p* = 0.050 and 0.057, respectively). ETA protein was slightly increased by cinacalcet alone (*p* = 0.028), whereas sitaxentan alone or in combination with cinacalcet did not influence ETA protein expression. The ETB:ETA protein ratio was increased by sitaxentan+cinacalcet (*p* = 0.006), while the effects of sitaxentan or cinacalcet alone were not significant (*p* = 0.060 and *p* = 0.091, respectively). However, these results must be interpreted with caution, as the data distributions were not completely normalized even after logarithmic transformations, except in the case of ETA protein, and all groups did not have quite equal variances.

## 3. Discussion

Our results show that both experimental surgical 5/6 nephrectomy and the adenine model of CRI are associated with a suppressed ratio of kidney ETB:ETA protein. Of note, 5/6 nephrectomy resulted in proteinuria and, especially in the extended model, significantly elevated BP, whereas adenine-induced interstitial nephritis increased neither urine protein excretion nor BP. Hyperuricemia induced a small increase in ETB protein expression, whereas no changes were observed in the ET-receptor expressions in response to the modifications of the calcium-phosphate metabolism. In separate two-way ANOVA analysis including only those rats that were treated with 0.3% adenine, combined sitaxentan and cinacalcet treatment slightly increased ETB protein and the ETB:ETA protein ratio, while cinacalcet alone slightly increased ETA protein in the kidney. The aforementioned interventions have had diverse effects on other CRI-related disturbances, including blood pressure, calcium-phosphate-vitamin D metabolism, renal damage and expression of components of RAS, that we have reported previously in more detail [17,30,31,32,33,34,35,36].

The present 5/6 nephrectomy model was associated with unfavorable changes in ET-receptor protein expressions, especially in the renal cortex; although ETB protein expression remained unchanged, ETA protein was up-regulated, and the ratio between ETB versus ETA protein expression was reduced in our two non-related experimental 5/6 nephrectomy studies. There were also similar changes in ET-receptor protein expressions in the renal medulla, although statistically significant p-values were only reached in the 12-week 5/6 nephrectomy model. In the 27-week model, reduced statistical power due to the loss of experimental animals may partly explain why the changes in the medulla did not reach statistical significance. Based on the known effects of ETA stimulation, the offset balance between ET-receptor expressions may result in a pro-inflammatory, pro-fibrotic and vasoconstrictive milieu [2,8] and further progression of CRI. The present results suggest possible links between renal tissue damage, reduced creatinine clearance, increased proteinuria, and unfavorable changes in the ETB:ETA protein ratio in the 5/6 nephrectomy model. The significance of the antiproliferative actions of ETB was denoted by a 12-week 5/6 nephrectomy study showing that selective ETA antagonist and RAS inhibition alone and in combination similarly reduced vascular smooth muscle cell (VSMC) proliferation in response to growth factors. In contrast, the proliferative response to tumor necrosis factor alpha (TNF-α) in rats receiving unselective ET-receptor antagonist treatment was increased when compared with the above treatment groups and untreated 5/6 nephrectomized rats [37].

The pathological reduction in the ETB versus ETA protein expression in renal tissue seems to be a rather early phenomenon in the course of experimental CRI, as such changes were already present in the 12-week 5/6 nephrectomy model and the changes were of a similar degree in the 27-week model, even though the latter presented with further reduction in creatinine clearance, pronounced proteinuria, and significantly elevated BP. Recently, the combining of a selective ETA antagonist to a RAS inhibitor treatment did not show any additional benefits in the well-established 5/6 nephrectomy model in renin transgenic hypertensive rats with up to 60 weeks of follow-up, while in such extreme models the addition of an ETA antagonist could even have unexpected negative effects [38].

The offset balance of ETB versus ETA receptor expressions was not modulated by changes in calcium-phosphate-vitamin-D metabolism. We previously found that the correction of the calcium-phosphate balance ameliorated renal and vascular pathophysiology in experimental CRI, possibly via suppressed renal and vascular RAS components [33,34,35]. There is a well-established link between the RAS and the ET-systems [5]. The present hyperuricemia had a minor increasing effect on ETB protein expression, but this was not reflected as changes in the ETB:ETA ratio in the kidney. Hyperuricemia has a contradictory role in the pathophysiology of CRI [39], and this controversy is supported by our previous findings. Previously, oxonic acid-induced hyperuricemia reduced oxidative stress and improved NO-mediated vasorelaxation in the carotid artery of 5/6 nephrectomized rats [31]. The antioxidant effect of uric acid might serve as one explanation for the small increase in ETB protein expression following oxonic acid treatment in the current study. In addition to the increased ETB protein in the kidney, we found that experimental hyperuricemia decreased proteinuria and improved tubulointerstitial morphology in the kidney. On the other hand, we previously found that experimental hyperuricemia increased plasma aldosterone [30], which has been suggested to decrease renal ETB activity via sulfenic acid modification and also to promote renal injury [20]. Furthermore, ET-1 is known to increase aldosterone production in the adrenal cortex via both ETB and ETA [40]. Future studies are warranted to establish the interplay between hyperuricemia, aldosterone and the ET-system in the pathophysiology of CRI.

The protein expressions of ETB and ETA in our adenine model of interstitial nephritis were both significantly reduced, which may have resulted from a lack of sufficient protein synthesis in the damaged renal tissue. We have previously shown the severe morphological changes induced by this model [17]. The difference in ETA protein synthesis between the 5/6 nephrectomy and adenine models may also have resulted from the different characteristics between the models; i.e., the presence of proteinuria and elevated BP in the 5/6 nephrectomy model. Both proteinuria and renal damage have been linked in a vicious-circle–like manner to pathological ET-system activation, while treatment with ETA antagonists has been shown to have antiproteinuric effects [12]. 

The reduced ratio between the ETB and ETA protein expressions in the adenine model also suggests that the offset ET-receptor balance is a potential pathophysiological mechanism also in CKD of interstitial origin without proteinuria. This corresponds to the known ETA-mediated mechanisms that can account for the progression of interstitial renal damage including vasoconstriction, coagulation, inflammation, extracellurar matrix accumulation, and fibrosis [2,8]. We previously reported that the adenine model was characterized by interstitial inflammation and fibrosis, tubular dilatation and atrophy, but only a low degree of glomerular injury [17]. In the previous study, sitaxentan improved creatinine clearance and reduced tubular atrophy and, in combination with cinacalcet, also alleviated interstitial inflammation and tubular dilatation. Taken together, the adenine model of interstitial renal damage is associated with an unfavorable change in the ETB:ETA ratio, and this imbalance and the adenine-induced kidney tissue damage can at least partially be counteracted by the selective ETA receptor antagonist sitaxentan and its combination with the calcimimetic cinacalcet [17]. Our current study also indicates a lack of any remarkable feed-back loop between ETA receptor activity and ET-receptor expression, as sitaxentan did not change the ET-receptor mRNA or protein expressions.

Although the present changes in the ET-receptor expressions by sitaxentan and cinacalcet were minute, the results of two-way ANOVA suggest an increase in ETB and the ETB:ETA ratio caused by the combination treatment [17]. Cinacalcet alone slightly increased kidney ETA protein, but the net effect on ETB:ETA ratio was beneficial. Previously, cinacalcet and the consequent reduction in PTH were found to down-regulate the activity of endothelin-converting enzyme 1 (ECE-1), which reduced the circulating levels of ET-1 and lowered BP, despite a compensatory increase in the ECE-1 protein [41]. In another study concerning adenine-induced CRI, the moderate reduction of PTH from 441 to 327 pg/mL with cinacalcet attenuated the endothelial-to-mesenchymal transition in rat kidneys, one of the mechanisms of myofibroblast accumulation in renal fibrogenesis [42]. The modulation of the calcium-phosphate and PTH levels in our twenty-seven-week 5/6 nephrectomy study had no effect on the ET-receptors, although the calcium diet improved tubulointerstital morphology. Future studies about the potential effects of cinacalcet and calcium-phosphate balance on the ET-system seem warranted.

The balance between ETB versus ETA has been previously studied in different forms of experimental CRI, but most of the studies have only examined the mRNA of the ET-receptors and the results have been contradictory. The ET-receptor expressions have been studied in surgical and ligation 5/6 nephrectomy models [21,27], in renal ischemia and ischemia-reperfusion injury models [16,19,20,29], in streptozotocin-induced diabetes [22,23], in a rat model of acute liver and renal failure induced by galactosamine [18], in a NZB/W F1 mouse model of lupus erythematosus and in polycystic kidneys of cpk mice [24,25], in puromycinaminonucleoside (PAN)-induced nephrosis [26], and in a rat model of mesangial proliferative glomerulonephritis induced by anti-thymocyte serum [28]. In accordance with our results, renal ischemia and ischemia-reperfusion injury models have induced changes in the ET-receptor expressions. The receptor expressions either on mRNA or protein levels, or even the receptor functions by post-translational modification, as well as the presence of NO deficiency, showed changes towards ETA dominancy in different rodent experiments [16,19,20,29]. Altogether, the ET-receptor mRNA expressions seem labile at different time points of CRI and the ET-receptor protein expressions seem to undergo significant modulation at the level of translation or thereafter [21,27]. Our study emphasizes the importance of investigating the parallel expressions at the protein level.

Clinical studies seem warranted in the future to establish the renal protein expressions of ETB and ETA in different forms of CKD. Contrary to our results, increased ET-1 and ETB receptor mRNA and protein expressions were demonstrated in a small study containing renal patients with different diseases characterized by high-grade proteinuria [43]. Studies evaluating potential CKD-related changes in the ET-receptor expressions in other tissues than kidney would also be of interest. The presence of such changes is supported by a study where the selective ETA antagonist atrasentan reduced medial calcifications and vascular stiffness in 5/6 nephrectomized rats that were given high doses of Ca+Pi and vitamin D [44]. In addition, atrasentan improved cardiac histology in 5/6 nephrectomized rats despite the renoprotective effects of the ETA antagonist being absent in this study [45]. The ET-receptor expressions were not reported in either of the aforementioned studies. With a remarkable analogy to our 5/6 nephrectomy studies, increased local expression of ETA receptor and unchanged ETB receptor protein expression have previously been reported in human aortic valve stenosis [46]. Local changes in favor of ETA receptor protein expression have also been reported in the myocardium of patients with chronic heart failure due to dilated cardiomyopathy [47].

In conclusion, our results show that both the 5/6 nephrectomy model and the adenine model of CRI are characterized by reduced ETB versus ETA protein expression, although the former is a model with high proteinuria and elevated BP, while the latter mainly affects the tubulointerstitial compartment of renal tissue in the absence of hypertension. Consequently, the offset balance between ETB versus ETA protein expressions may be an important pathophysiological mechanism in CRI irrespective of the initial renal insult or the presence of proteinuria. ETA receptor antagonists and ETB mimetics may therefore also be beneficial in CKD without proteinuria and should be tested in a wider spectrum of renal diseases in the future.

## 4. Materials and Methods

### 4.1. Animals and Experimental Design

The experimental designs of all three CRI models were approved by the Animal Experimentation Committee of the University of Tampere and the Provincial Government of Western Finland, Department of Social Affairs and Health, Finland (decision numbers: LSLH-2003-9718/Ym-23, 4th of December 2003; ESAVI-2010-03090/Ym-23, 26th of January 2010). The study protocols conformed to the Guiding Principles for Research Involving Animals.

#### 4.1.1. Twelve-Week 5/6 Nephrectomy Model with Oxonic Acid Diet Modulation

Male Sprague–Dawley rats (*n* = 48) were housed two to a cage with free access to water and food (Lactamin R34, AnalyCen, Lindköping, Sweden). At the age of 8 weeks, rats underwent the NX or sham operation. NX was performed by surgical resection of the upper and lower poles (2/3) of the left kidney, followed by contralateral nephrectomy. Anesthesia, antibiotics, postoperative pain relief and systolic BP measurement by tail cuff were performed as described in previous reports [30,31,32].

Three weeks after the operation, both the NX and sham rats were divided into two groups with equal systolic BPs and body weights. The other half of both NX and sham rats was switched to chow containing 2.0% oxonic acid for 9 weeks while the other halves continued on the control diet. The groups were, consequently, Sham, Sham+oxonic acid diet (Oxo), NX, and NX+oxonic acid diet (NX+Oxo). The animal number in each group was 12. Due to cardiac arrest during anesthesia at the close of the study, blood samples could not be collected from one NX and three Sham rats. Otherwise there was no premature attrition in this study.

During the final study week, 24-h fluid consumption and urine output were measured in metabolic cages. Then, the rats were anesthetized (urethane 1.3 g/kg) and blood samples were drawn from the cannulated carotid artery with EDTA or heparin being used as anticoagulants, as appropriate. The kidneys were removed and weighed. A sagittally cut half of kidney sample from each rat was snap-frozen in isopentane at –40 °C and stored at −80°C for mRNA and protein determinations.

#### 4.1.2. Twenty-Seven-Week 5/6 Nephrectomy Model with Calcium and Phosphate Diets and Intraperitoneal Paricacitol Injections

Male Sprague–Dawley rats (*n* = 64) underwent identical study protocol as described in the first and last paragraphs of the 12-week NX model above. Fifteen weeks after the operations, the NX rats were divided into four groups with similar systolic BPs, body weights and plasma creatinine values. For the following 12 weeks, the diets were modified: the Sham and NX groups were given 0.3% Ca + 0.5% Pi, the NX+Ca group 3.0% Ca + 0.5% Pi and the NX+Pi group 0.3% Ca + 1.5% Pi. The diets represent intakes of low Ca/moderate Pi, high Ca/moderate Pi, and low Ca/high Pi, respectively. The calcium diets were based on our previous studies of 0.3% versus 3.0% calcium intake in experimental CRI [33,34,35]. In addition, N+Pari group received 0.3% Ca + 0.5% Pi combined with intraperitoneal paricalcitol injections (100 ng/rat three times a week). During the intervention period, the body weight and systolic BPs were measured every two weeks. Due to premature loss of 6, 1, 6 and 4 rats in the NX, NX+Ca, NX+Pi and NX+Pari groups, respectively, the final animal numbers in these groups were 7, 11, 7 and 9, respectively, and in the Sham group 13.

#### 4.1.3. Twelve-Week Adenine Model with Endothelin a Receptor Antagonist and Calcimimetic

Male Wistar rats (*n* = 80) were housed four to a cage with free access to water and chow (RM3, Scanbur, Karlslunde, Denmark). At the age of 10 weeks, the rats were divided into 5 groups with equal systolic BPs and body weights. One group was given the control diet and 4 groups received 0.3% adenine (Sigma-Aldrich, Saint Louis, MO, USA) added to the chow for the following 12 weeks. In addition to adenine, the rats in 3 groups received either sitaxentan (50 mg/kg/d in the drinking water; Pfizer, New York, NY, USA), cinacalcet (20 mg/kg/d added to the 0.3% adenine chow; Amgen Thousand Oaks, CA, USA), or both of these treatments for 12 weeks. The study groups were: control (Control, *n* = 12), adenine (Ade, *n* = 20), adenine+sitaxentan (Ade+S, *n* = 16), adenine+cinacalcet (Ade+C, *n* = 16), and adenine+combination treatment (Ade+SC, *n* = 16). The medications continued until the end of the study.

Body weights were monitored weekly, systolic BP was measured at the end of the study, and 24-h water consumption, urine output, and chow consumption were monitored twice during the treatments (weeks 5 and 10) in metabolic cages. At the close of the study, the rats were anesthetized (urethane 1.3 g/kg) and blood samples were drawn from the cannulated carotid artery with EDTA and heparin as anticoagulants, as appropriate. The urine, plasma and weighed tissue samples were stored at −70°C [17].

### 4.2. Plasma and Urine Determinations

All biochemical measurements were performed as previously described [17,30,31,32,33,34,35]. The determination of creatinine clearance was based on 24-h urine collection and plasma samples taken at the close of the studies.

### 4.3. Real-Time Quantitative RT-PCR

The total RNA was isolated from kidney tissue using Trizol reagent (Invitrogen, Carlsbad, CA, USA). The reverse transcription of the RNA was carried out using M-MLV reverse transcriptase (Invitrogen) according to the manufacturer’s instructions. Beta-actin was used as a housekeeping gene. The PCRs were performed with SYBR Green or TaqMan chemistry using ABI PRISM 7000 sequence detection (Applied Biosystems, Foster City, CA, USA).

The PCRs for ETB and ETA were performed in duplicate in 25 μL final volume containing 1X SYBR Green Master mix (Applied Biosystems) and 300 nM of primers. PCRs for Beta-actin were performed in duplicate in 25 μL final volume containing 1X TaqMan Master mix (Applied Biosystems) and 1X Beta-actin TaqMan Gene Expression Assay (TaqMan assay code Rn00667869_m1, Applied Biosystems). The PCR cycling conditions for the mRNAs were 10 min at +95°C and 40 cycles of 20 s at +95 °C and 1 min at +60 °C. The data were analyzed using the absolute standard curve method [48]. The expression of Beta-actin did not differ between the groups, allowing its use as a control gene.

### 4.4. Kidney Autoradiography

For autoradiographic studies, frozen kidney sections (20 µm thick) were cut on a cryostat at −17 °C, thaw mounted onto Super Frost Plus slides (Menzel-Gläser, Germany), dried in a desiccator under reduced pressure at 4 °C overnight, and stored at −80°C with silica gel until further processing. Quantitative in vitro autoradiography of ETB and ETA receptors was performed with the radioligand [125I]-ET-1 (Peptide Institute Inc., Osaka, Japan) as described previously [33,35,49,50]. The density of ETB was determined in the presence of the ETA receptor selective antagonist BQ123, and the density of ETA in the presence of the ETB receptor selective agonist Sarafotoxin-6c (S6c).The optical densities were quantified by an image analyzing system (AIDA 2D densitometry) coupled to the FUJIFILM BAS-TP2025 phosphorimager (Tamro, Finland).

### 4.5. ELISA

The kidney tissues were homogenized in a lysis buffer containing 100 mmol/L NaCl, 10 mmol/L KCl, 8 mmol/L Na2HPO4, 3 mmol/L MgCl2, 0.5% NP40, 10 mmol/L Tris–HCl, pH 7.4 and protease inhibitors (Complete^TM^ Mini, Protease Inhibitor Cocktail Tablets, Roche Diagnostics GmbH, Mannheim, Germany). The homogenates were incubated on ice for 30 min and centrifuged at 15,000 *g* for 15 min. The protein concentration in the supernatant was determined using the BCA protein assay kit (Pierce, Rockford, IL, USA). The ETA and ETB protein levels were measured using ELISA (MyBioSource Rat Endothelin A receptor Kit MBS7606745 and Rat Endothelin B receptor ELISA Kit MBS7606746), respectively, according to the manufacturer’s instructions. The results are expressed as ng of ETA and ETB protein in μg of total protein.

### 4.6. Tubulointerstitial Kidney Histology

In the 5/6 nephrectomy studies, five-mm-thick kidney sections were stained with hematoxylin-eosin (HE) or periodic acid-Schiff (PAS) and processed for light microscopic evaluation. The histology analyses were performed while blinded to the treatments. Tubulointerstitial damage (HE and PAS stain) was scored (from 0 to 4) as follows: tubular atrophy, dilation, casts, interstitial inflammation, and fibrosis were assessed in 10 kidney fields at a magnification of X 100: 0 = normal, 1 = lesions in < 25% of the area, 2 = lesions in 25−50% of the area, 3 = lesions in > 50% of the area, 4 = lesions involving the entire area [35,51].

### 4.7. Data Presentation and Analysis of Results

The statistical analyses in the 12-week 5/6 nephrectomy study were carried out using two-way ANOVA for the variables that were normally distributed and had equal variances verified by visual inspection and the Kolmogorov–Smirnov test as well as by Levene’s test, respectively. Logarithmic transformation was used for variables that were not normally distributed. If the variables were not normally distributed and/or the variances were not equal, Kruskal–Wallis test and Mann–Whitney U-test were used instead.

Statistical analyses in the 27-week 5/6 nephrectomy study and 12-week adenine study were carried out using one-way ANOVA supported by a post-hoc Bonferroni test or a Kruskal–Wallis test and Mann–Whitney U-test with Bonferroni correction. The selection between parametric and nonparametric test was made as described above. Logarithmic transformation was used to normalize the distribution of the variables as appropriate. Additionally, we analyzed the rat groups receiving adenine using two-way ANOVA, although all of the variables were not completely normalized even after logarithmic transformation.

The results in all tables are presented as mean ± SEM, whereas the figures show mean ± 95% confidence interval of the mean. Differences were considered significant when *P*<0.05. SPSS 25.0 software (IBM SPSS Statistics, Armonk, NY, USA) was used for the statistics.

## Figures and Tables

**Figure 1 ijms-21-00936-f001:**
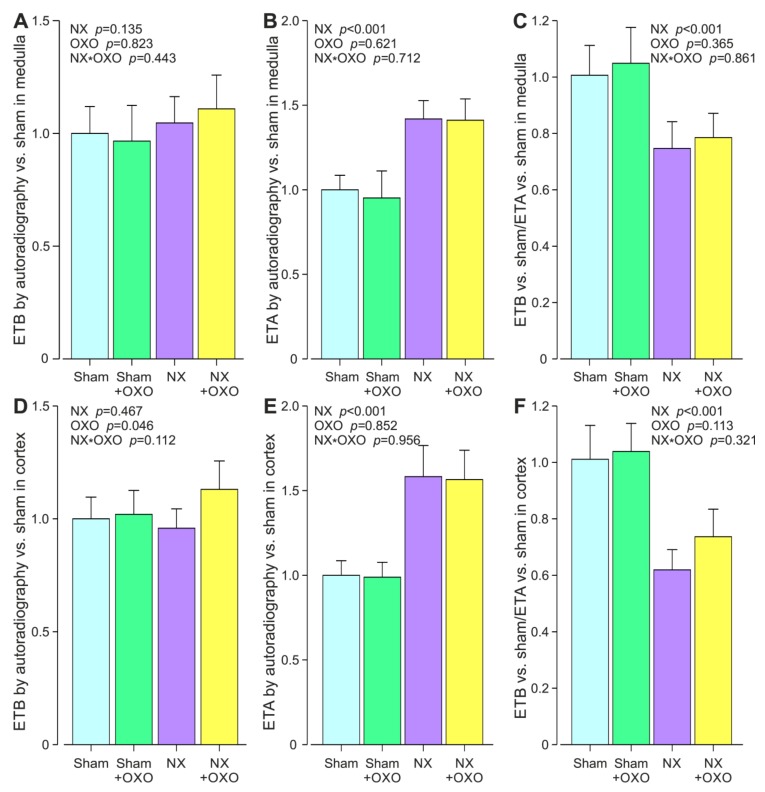
Kidney medullary ETB (**A**) and ETA (**B**) receptor protein content, and the ETB:ETA protein ratio (**C**); and kidney cortical ETB (**D**) and ETA (**E**) receptor protein content, and the ETB:ETA receptor protein ratio (**F**) quantified using autoradiography in the twelve-week 5/6 nephrectomy model; NX, 5/6 nephrectomy; Sham, sham-operation; OXO, 2.0% oxonic acid diet; *n* = 12/group. Values are mean ± 95% confidence interval of the mean. Two-way ANOVA *p*-values for NX, OXO and their interaction are presented in the figures.

**Figure 2 ijms-21-00936-f002:**
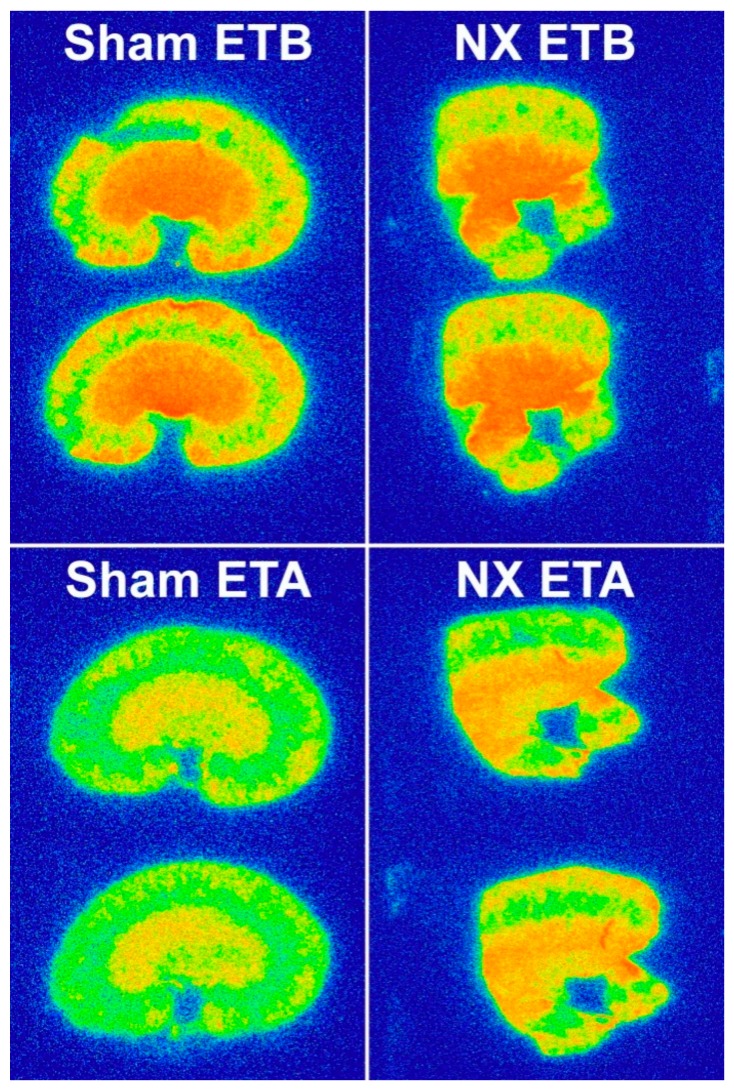
Representative original tracings of kidney ETB and ETA receptor autoradiography in the sham-operated and 5/6 nephrectomized rats in the twelve-week 5/6 nephrectomy model.

**Figure 3 ijms-21-00936-f003:**
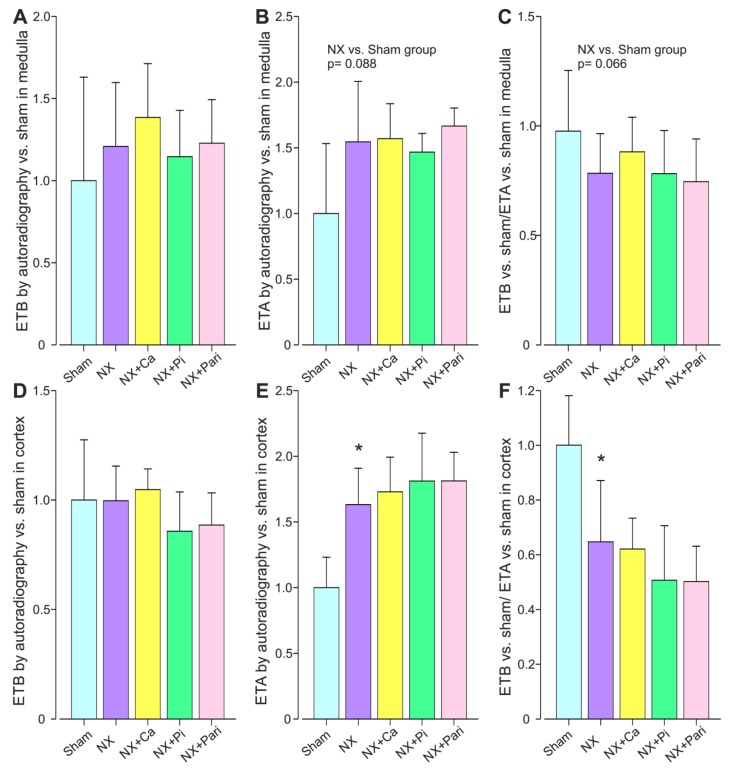
Kidney medullary ETB (**A**) and ETA (**B**) receptor protein content, and the ETB:ETA protein ratio (**C**); and kidney cortical ETB (**D**) and ETA (**E**) receptor protein content, and the ETB:ETA receptor protein ratio (**F**) quantified using autoradiography in the twenty-seven-week 5/6 nephrectomy model; NX, 5/6 nephrectomy; Sham, sham-operation; Ca, 3.0% calcium diet; Pi, 1.5% phosphate diet; Pari, 100 ng/rat of intraperitoneal paricalcitol three times weekly; *n* = 7−13/group. Values are mean ± 95% confidence interval of the mean. * *p* < 0.05 vs. Sham.

**Figure 4 ijms-21-00936-f004:**
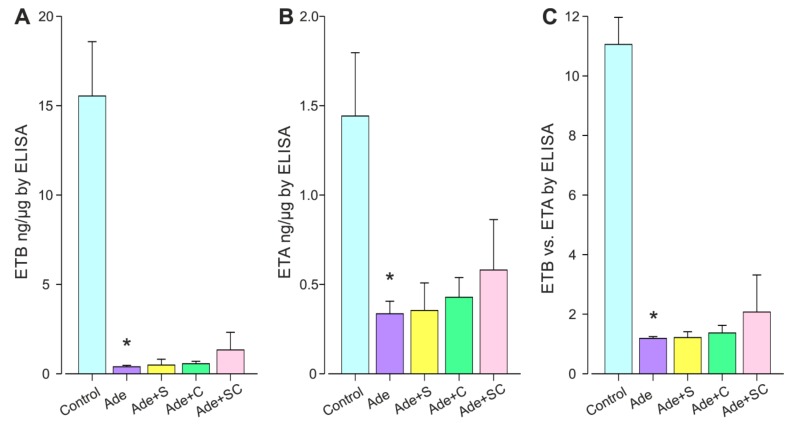
Kidney ETB (**A**) and ETA (**B**) receptor protein content, and the ETB:ETA receptor protein ratio (**C**) quantified using ELISA in the twelve-week adenine model; Control, control diet; Ade, 0.3% adenine diet; S, sitaxentan50 mg/kg/day; C, cinacalcet 20 mg/kg/day; SC, sitaxentan+cinacalcet; *n* = 12–20/group. Values are mean ± 95% confidence interval of the mean. * *p* < 0.05 vs. Control.

**Table 1 ijms-21-00936-t001:** Animal data and laboratory findings at the end of the twelve-week 5/6 nephrectomy model.

	Sham	Sham+Oxo	NX	NX + Oxo	Two-Way ANOVA *p*-Values		
					NX	Oxo	Interaction NX*Oxo
Number of animals	12	12	12	12			
Body weight (g) week 9	433 ± 8	412 ± 11	448 ± 10	411 ± 8	0.441	0.004	0.377
Systolic blood pressure mmHg week 9	134 ± 7	136 ± 5	142 ± 6	152 ± 4	0.041	0.290	0.498
Creatinineclearance (mL/min)	2.89 ± 0.36	1.99 ± 0.16	1.21 ± 0.06*	1.19 ± 0.09 †	ND	ND	ND
Plasma determinations	(*n* = 9)	(*n* = 12)	(*n* = 11)	(*n*= 12)			
Creatinine (µmol/L)	40.2 ± 5.1	48.5 ± 3.0	81.9 ± 3.1	83.1 ± 8.1	<0.001	0.204	0.128
Urea (mmol/L)	6.6 ± 0.3	8.3 ± 0.4	13.5 ± 0.9	14.5 ± 2.0	<0.001	0.099	0.171
Uric acid (µmol/L)	36.0 ± 10.7	117.3 ± 20.9 *	62.9 ± 19.0	152.3 ± 19.0 #	ND	ND	ND
Calcium (mmol/L)	2.42 ± 0.02	2.36 ± 0.02	2.43 ± 0.04	2.42 ± 0.02	0.280	0.235	0.399
Phosphate (mmol/L)	1.38 ± 0.07	1.51 ± 0.06	1.88 ± 0.17	1.92 ± 0.18	0.002	0.427	0.618
Urine protein (mg/24 h)	70.5 ± 10.1	44.0 ± 8.6	323.2 ± 31.6	247.0 ± 44.0	<0.001	0.006	0.616
Tubulointerstitial damage index	0.0 ± 0.0	0.0 ± 0.0	2.3 ± 0.4*	0.7 ± 0.3†#	ND	ND	ND
Kidney Endothelin receptors							
ETB mRNA copies × 10^3^/ng total RNA	4.02 ± 0.29	2.61 ± 0.39	0.97 ± 0.15	0.90 ± 0.13	<0.001	0.037	0.085
ETA mRNA copies × 10^2^/ng total RNA	2.49 ± 0.26	1.72 ± 0.26	0.70 ± 0.14	0.81 ± 0.18	<0.001	0.281	0.159
ETB vs. ETA mRNA	17.2 ± 1.1	16.7 ± 1.4	15.4 ± 1.6	15.7 ± 2.8	0.156	0.474	0.786

Values are means ± SEM. NX, 5/6 nephrectomy; Sham, sham-operation; Oxo, 2.0% oxonic acid diet. ETB = endothelin receptor B; ETA = endothelin receptor A. ND = not determined as the variable was not normally distributed and/or the variances were not equal, and the statistical analyses were made by Kruskal-Wallis and Bonferroni corrected Mann-Whitney U tests instead;* *p* < 0.05 Sham-Oxo or NX vs. Sham; † *p* < 0.05 NX+Oxo vs. Sham+Oxo; # *p*< 0.05 NX-Oxo vs. NX.

**Table 2 ijms-21-00936-t002:** Animal data and laboratory findings at the end of the twenty-seven-week 5/6 nephrectomy model.

	Sham	NX	NX+Ca	NX+Pi	NX+Pari
Number of animals	13	7	11	7	9
Body weight (g)	565 ± 8	507 ± 41	481 ± 15	431 ± 38	503 ± 32
Systolic blood pressure mmHg	130 ± 2	171 ± 5 *	143 ± 4 #	161 ± 4	167 ± 5
Creatinineclearance (mL/min)	1.84 ± 0.11	0.85 ± 0.17 *	0.84 ± 0.07	0.69 ± 0.18	0.71 ± 0.15
Plasma determinations					
Creatinine (µmol/L)	66.6 ± 2.2	170.0 ± 37.7 *	116.5 ± 7.3	178.0 ± 30.9	209.0 ± 41.7
Urea (mmol/L)	5.3 ± 0.2	23.3 ± 8.7 *	12.9 ± 1.1	34.9 ± 11.5	33.8 ± 9.8
25OH-D_3_ (nmol/L)	33.4 ± 2.8	19.8 ± 4.0 *	13.8 ± 0.9	16.0 ± 1.3	13.0 ± 0.9
1,25(OH)_2_D_3_ (pmol/L)	273.0 ± 27.8	70.5 ± 22.9 *	105.8 ± 22.0	105.5 ± 42.9	6.2 ± 1.2#
Ionized calcium (mmol/L)	1.35 ± 0.01	1.34 ± 0.03	1.59 ± 0.04 #	0.93 ± 0.09 #	1.31 ± 0.05
Phosphate (mmol/L)	1.19 ± 0.05	2.52 ± 0.54 *	0.70 ± 0.06 #	5.47 ± 1.21	3.03 ± 0.43
PTH (pg/mL)	102.4 ± 52.7	1172.8 ± 369.8 *	3.7 ± 0.5 #	3619.7 ± 255.0 #	618.9 ± 253.8
Urine protein (mg/24 h)	132.7 ± 21.0	359.1 ± 49.6 *	522.5 ± 50.6	626.8 ± 39.5 #	432.5 ± 56.6
Tubulointerstitial damage index	0.6 ± 0.1	2.9 ± 0.3 *	1.9 ± 0.1#	3.2 ± 0.3	3.5 ± 0.3
Kidney Endothelin receptors					
ETB mRNA copies × 10^4^/ng total RNA	0.84 ± 0.07	1.41 ± 0.10 *	1.52 ± 0.07	1.23 ± 0.10	1.28 ± 0.06
ETA mRNA copies × 10^3^/ng total RNA	0.93 ± 0.08	2.43 ± 0.39 *	2.12 ± 0.11	2.38 ± 0.29	1.85 ± 0.19
ETB vs. ETA mRNA	9.9 ± 1.4	6.7 ± 1.4	7.5 ± 0.8	5.7 ± 1.0	8.3 ± 2.0

Values are means ± SEM. * *p* < 0.05 NX vs. sham; # *p* < 0.05 NX+Ca, NX+Pi or NX+Pari vs. NX. NX, 5/6 nephrectomy; Sham, sham-operation; Ca, 3.0% calcium diet; Pi, 1.5% phosphate diet; Pari, 100 ng/rat of intraperitoneal paricalcitol three times weekly. PTH = parathyroid hormone; ETB = endothelin receptor B; ETA = endothelin receptor A.

**Table 3 ijms-21-00936-t003:** Animal data and laboratory findings at the end of the twelve-week adenine model.

	Control	Ade	Ade+S	Ade+C	Ade+SC	Adenine Groups Two-Way ANOVA *p*-Values		
						Sita	Cina	Interaction Sita*Cina
Number of animals	12	20	16	16	16			
Body weight (g)								
week 12	450 ± 11	339 ± 6 *	341 ± 6	340 ± 7	359 ± 8	0.136	0.160	0.253
Systolic Blood Pressure (mmHg)								
week 9	137 ± 2	138 ± 1	130 ± 2 †	138 ± 1 ‡	129 ± 2	<0.001	0.624	0.722
Creatinine clearance (mL/min)	1.70 ± 0.12	0.28 ± 0.02 *	0.41 ± 0.03 †	0.33 ± 0.04	0.44 ± 0.05	<0.001	0.494	0.743
Plasma determinations								
Creatinine (µmol/l)	61.3 ± 4.0	212.6 ± 9.0 *	160.2 ± 9.2 †	202.4 ± 16.6	154.2 ± 10.2	<0.001	0.309	0.794
Urea (mmol/L)	9.0 ± 0.4	40.8 ± 1.4 *	32.5 ± 1.3 †	39.8 ± 2.2	30.8 ± 1.8	<0.001	0.270	0.728
Uric acid (µmol/l)	93.8 ± 12.0	30.1 ± 4.6 *	18.6 ± 1.4 †	31.7 ± 4.5 ‡	18.0 ± 1.5	<0.001	0.915	0.650
Calcium (mmol/L)	2.31 ± 0.02	2.27 ± 0.05	2.37 ± 0.12 ‡	2.20 ± 0.03	2.10 ± 0.03	ND	ND	ND
Phosphate (mmol/L)	1.10 ± 0.09	1.83 ± 0.09 *	1.53 ± 0.08 ‡	1.95 ± 0.12	2.06 ± 0.08	ND	ND	ND
PTH (pg/mL)	31.6 ± 6.6	558.2 ± 51.2 *	324.7 ± 63.1 †,‡	89.6 ± 28.2 †	47.9 ± 9.4	ND	ND	ND
Urine protein (mg/24 h)	14.7 ± 1.1	13.9 ± 1.7	11.2 ± 1.1	12.5 ± 2.6	11.7 ± 1.4	0.710	0.522	0.416
Kidney Endothelin receptors								
ETB mRNA copies × 104/ng total RNA	2.69 ± 0.17	1.70 ± 0.10 *	1.37 ± 0.12	1.67 ± 0.13	1.51 ± 0.10	ND	ND	ND
ETA mRNA copies × 103/ng total RNA	1.22 ± 0.13	5.78 ± 0.34 *	5.41 ± 0.51	5.50 ± 0.43	4.47 ± 0.34	0.090	0.137	0.417
ETB vs. ETA mRNA	23.8 ± 1.9	3.0 ± 0.2 *	2.6 ± 0.2	3.1 ± 0.2	3.7 ± 0.4	0.715	0.034	0.080

Values are means ± SEM. Primary group-wise comparisons: * *p* < 0.05 Ade vs. Control; † *p* < 0.05 Ade+S or Ade+C vs. Ade; ‡ *p* < 0.05 Ade+S or Ade+C vs. Ade+SC. Secondary two-way ANOVA analyses in the adenine groups only are shown in the final three columns of the table; ND = not determined as the variable was not normally distributed and/or the variances were not equal. Control, control diet; Ade, 0.3% adenine diet; S, sitaxentan 50 mg/kg/day; C, cinacalcet 20 mg/kg/day; SC, sitaxentan+cinacalcet. Sita, sitaxentan effect in the adenine groups; Cina, cinacalcet effect in the adenine groups. PTH = parathyroid hormone; ETB = endothelin receptor B; ETA = endothelin receptor A.

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
