# Peer review of "Unfavorable Reduction in the Ratio of Endothelin B to A Receptors in Experimental 5/6 Nephrectomy and Adenine Models of Chronic Renal Insufficiency"

_ijms, 2020, doi:10.3390/ijms21030936_

Round 1
Reviewer 1 Report
The authors cleared issues that I raised, and thus this manuscript meets the standards for publication in IJMS.
Author Response
Reviewer 1:
The authors cleared issues that I raised, and thus this manuscript meets the standards for publication in IJMS.
Response:
The authors wish to thank the reviewer for the thoughtful comments and advice for the revision of the paper. We were pleased to hear that both reviewers thought our manuscript had improved a lot.
Reviewer 2 Report
The revised paper is much improved.
In Table 2 the single dagger indicating a reduction in systolic pressure in the NX+Ca group should be replaced by a hatch (#).
In Table 3 you can also include columns showing the P values of the 2-way ANOVA (as was done in Table 1).
Author Response
Reviewer 2:
The revised paper is much improved.
In Table 2 the single dagger indicating a reduction in systolic pressure in the NX+Ca group should be replaced by a hatch (#). In Table 3 you can also include columns showing the P values of the 2-way ANOVA (as was done in Table 1).
Response:
We thank the reviewer for the professional and constructive comments that have helped us to improve our manuscript. All corrections to our manuscript suggested by the Reviewer 2 have been made in blue text font as follows:
As carefully noticed by the Reviewer 2, we have replaced the single dagger by a hatch in Table 2 in the systolic blood pressure of the NX-Ca group. We also added the p-values of the separate 2-way ANOVA analyses in the adenine groups in the final three columns of Table 3, as suggested by the Reviewer 2. However, as we mentioned in our previous response letter, many variables in the adenine model were not normally distributed even after logarithmic transformation (log10 or ln) and/or did not have equal variances between the groups. In these cases, we left the two-way ANOVA p-values not determined in Table 3 as the assumptions of the parametric two-way ANOVA test were not fulfilled. The separate two-way ANOVA analyses in the adenine groups did not reveal any major effects compared to our primary analyses between the groups. A short description of the two-way ANOVA results in Table 3 has been added to the Results section (lines 219-221).
This manuscript is a resubmission of an earlier submission. The following is a list of the peer review reports and author responses from that submission.
Round 1
Reviewer 1 Report
The subject of this study and experimental methods are interesting. And the result of this study can expand our knowledge of renal injury and pathophysiology. I have just one question as follows.
In figure 4B and 4E, the mRNA level of ETA was increased but its protein level was significantly decreased. How can you explain this discrepancy between mRNA and protein levels of ETA? In addition, while the ETA expression at the level of protein is decreased, how is the endothelin system activated and how is sitaxentan effective to ameliorate renal injury?
Author Response
In figure 4B and 4E, the mRNA level of ETA was increased but its protein level was significantly decreased. How can you explain this discrepancy between mRNA and protein levels of ETA?
response: We agree that the discrepancy between mRNA and protein levels of the ETreceptors was evident in this study. Based on the results of the two 5/6 nephrectomy studies, the mRNA expression seemed labile between different time points, contrary to the corresponding changes of protein levels in both 12-week and 27-week 5/6 nephrectomy study. The incoherence of the ET-receptor mRNA expressions is supported by the contradictory findings of previous studies concerning ET-receptor mRNA levels (references 18-29 in the present manuscript).
As shown well by the poor correlation between the mRNA and protein results in our study, the expression of various genes should not only be examined using mRNA determinations, but the determinations should be confirmed at the
protein level. This is because the mRNA levels give no information about the functional level of the gene product, i.e. the level of translation as well as the regulation of longevity of the end-product thereafter. We have removed the
mRNA results from the figures to the tables as suggested by the other reviewer.
In addition, while the ETA expression at the level of protein is decreased, how is the endothelin system activated and how is sitaxentan effective to ameliorate renal injury?
response: In the adenine model ETA protein expression was decreased contrary to the raised ETA protein levels in the 5/6 nephrectomy models.The discrepancy in ETA protein expressions between the models might relate to the different models themselves and the presence of proteinuria in the 5/6 nephrectomy model. We previously published in more detail that the adenine model was characterised by severe tubulointerstital damage and atrophied and dilated tubules (reference 17 in the present manuscript). The severe morphological changes might partly explain the reduced protein synthesis in the model as stated on lines 293-295. Importantly, although both ETB and ETA protein were decreased in the adenine model, ETB protein was decreased clearly more than ETA, and the ETB:ETA ratio was consequently reduced in the adenine model like in the 5/6 nephrectomy models. Therefore, both the 5/6 nephrectomy model and the adenine model were characterised by an unfavourable offset in the balance between the opposing ETB and ETA receptors.
In the current study, we focused on the receptor expressions and did not measure ET-1 production as it has been previously well-established that CKD is featured by increased systemic and renal production of ET-1 regardless of the
underlying cause (reference 2 in the present manuscript). We avoided mentioning “ET-system activation” (referring to raised production or activity of all ET-system components) in the discussion for the aforementioned reasons.
Instead, we emphasised that our results showed the offset balance between the ETB and ETA receptor expression towards harmful ETA dominancy. In addition to the previously shown rise in ET-1, our findings support a pathophysiological mechanism relevant to CRI of different origins.
Reviewer 2 Report
This is an extensive study on the expression of endothelin (ET) A and B receptors in the kidney cortex and medulla of proteinuric 5/6 nephrectomized (NX) rats at two different time points: 12 and 27 weeks after surgery in combination with diverse interventions and in the non-proteinuric dietary adenine tubulo-interstitial injury model after 12 weeks of adenine exposure with a third set of interventions. This mixed bag makes it very difficult to extract the main message from the data set. The omission of a clear hypothesis in the introduction suggests that the authors also had difficulties in finding a main message. The statistics are not appropriate for the design and need to be redone.
I have a number of suggestions that should help to improve data presentation and direct the focus of the reader to the most important findings.
Major comments
A) The most interesting data are the quantitative autoradiography data of cortical and medullary ETA and ETB protein expression (and their ratio) in 5/6NX vs. sham rats at 12 and 27 weeks and the quantitative ELISA of whole kidney in Adenine vs. controls rats at 12 weeks. At both 12 and 27 weeks after 5/6NX cortical ETA expression is up and cortical ETB expression is down resulting in an unfavorable ratio. In the medulla this is not yet apparent at 12 weeks but also the case at 27 weeks. In contrast, in the adenine model whole kidney ETA and ETB expression are both markedly reduced. However, because the reduction in ETB is relatively stronger the ratio also becomes unfavorable. This combination of findings leads me to the main conclusion, namely that chronic renal injury, irrespective of either the initial insult or the presence of proteinuria, is characterized by this unfavorable ratio of ET receptor subtype expression. This should be formulated as the main hypothesis. The data support this hypothesis and thus suggest that there is a potential therapeutic opportunity for ETA antagonists and/or ETB mimetics in a broad range of subjects with CKD. To support this Figures 1, 3 an 4 should be devoted ETA and ETB protein expression (and their ratio) only and the mRNA data should be shifted to the tables. The medullary ETA and ETB protein expression (and the ratio) should be moved from Tables 1 and 2 to Figures 1 and 3. Because ETA and ETB mRNA expression (and their ratio) change inconsistently, as has also been previously documented by other groups, less emphasis should be placed on these data and the discussion on mRNA expression should be shortened and condensed.
B) The experiment with oxonic acid in 5/6NX vs. sham rats at 12 weeks has a symmetrical 2x2 design. This powerful design should be analyzed by a 2-way ANOVA allowing identification of effects related to 5/6NX, oxonic acid and their interaction. In the post-hoc analysis no comparison should be made between the 5/6NX+oxonic acid and sham groups. The 5/6NX+oxonic acid group should be compared to the sham+oxonic acid group and the 5/6NX+control diet groups only. The conclusion from this experiment is straightforward: hyperuricemia has no effect on renal ET expression (protein or mRNA) in either healthy or CKD rats. This should be (briefly) discussed in relation to relevant literature. It would be useful to also document tubulo-interstitial injury and inflammation in these groups to see whether any morphological changes can be identified that do not appear to impact ET expression.
C) The experiments with different Ca/P diets in the 5/6NX rats at 27 weeks also show little effect on ET expression. In this case a 1-way ANOVA is acceptable but the reference group should be the 5/6NX+control diet group and NOT the sham group! Thus the sham group should only be compared with the 5/6NX+control diet group and not with the intervention groups. The conclusion from this experiment is straightforward: the Ca/P/vit D axis does not appear to affect renal ET expression (protein or mRNA) in CKD rats. This should be (briefly) discussed in relation to relevant literature. Once again, it would be useful to also document tubulo-interstitial injury and inflammation in these groups to see whether any morphological changes can be identified that do not appear to impact ET expression.
D) For the adenine experiment I would choose to combine two statistical approaches: primarily a direct comparison (t-test) between the Adenine group and the control group to support the main conclusion of the whole study (see comment #1). A secondary analysis would be a 2-way ANOVA of the Adenine (Ade), Ade+sitaxentan (Ade+S), Ade+Cinacalcet (Ade+C) and the combined treatment (Ade+SC) groups. Once again, in the post-hoc analysis no comparison should be made between the Ade+SC and Ade groups. The Ade+SC group should be compared to the Ade+S and Ade+C groups only. This approach may actually reveal an effect of these interventions on ET expression in Ade model that is now masked by the huge depression of ET expression induced by Adenine as such. This would fit other recent reports, for instance (extrarenal) protective effects of an ETA blocker in the 5/6NX model on vascular calcification (Larivière et al. J Hypertens 2017) and cardiac remodeling (Ritter et al. Kidney Blood Press Res 2014). The findings should also be discussed in relation to the previously published renal morphology from this sub-study (reference #16).
Minor comments
E) The number of rats in each group shown in Table 1 is 12. Please indicate the number of plasma determination between brackets (9), (12), (11) and (12) in the plasma determination row.
F) Statistics: Proteinuria data in the 5/6NX model are not normally distributed (Tables 1 and 2). The data should be log-transformed prior to statistical analysis. This also applies to 1,25(OH2)D3 in Table 2 and PTH in Tables 2 and 3. Normality of data distribution should be checked for every variable. This is calculated as part of the ANOVA by most programs and therefore should not (only) be assessed visually. Log-transformation of data is always preferable to a non-parametric analysis.
G) The first paragraph of the introduction opens with the statement that ET receptor antagonists may have a therapeutic effect on top of RAS blockade. First of all this clearly only applies to ETA blockers and not to mixed ET blockers, and secondly in combination with other drugs such as RAS blockade plus soluble epoxide hydrolase blockade, ETA blockade can have unexpected effects (Čertíková Chábová et al. Kidney Blood Press Res 2019). A recent review summarizes ETA blocker studies in CKD models (Vaněčková et al. Physiol Res 2018).
Author response
A) To emphasise more the main hypothesis of the study, i.e. the CRI-related unfavorable changes in the ratio between ETB and ETA expression, we included the following sentence at the end of the introduction (lines 87-90): We tested the hypothesis whether there are changes in the ETB or ETA expressions or the ETB:ETA ratio in these experimental models of kidney damage, which would suggest that these mechanisms contribute to the pathophysiology of CRI
irrespective of the original cause of renal damage.
We completely agree with the reviewer about the main conclusion of the study, i.e. the unfavourable change in the ratio between ETB and ETA protein expression in two models of CRI and the consequent therapeutic potential of ETA antagonists or ETB mimetics in a wide range of CKD. In the final
paragraph of the discussion we also included ETB mimetics as a potential therapeutic approach in addition to ETA antagonists, as suggested by the reviewer. In accordance with the suggestions of the reviewer, we changed the
autoradiography results of the ETB and ETA protein expression in medulla to the figures 1 and 3 and moved the mRNA results to the tables 1, 2 and 3. We also shortened the discussion about the mRNA results.
B) As instructed by the reviewer, and in guidance of our professional statistician MSc Heini Huhtala, we re-analysed the 12-week 5/6 nephrectomy study using 2-way ANOVA. As also suggested by the reviewer, we tested the normality of all variables with Kolmogorov-Smirnov test instead of mere visual histogram analysis and performed logarithmic transformations (log10 or ln) to the variables that were not normally distributed. However, the distribution of some variables was not normalised even after the logarithmic transformations and/or the variances were not equal. Therefore, we were still compelled to use nonparametric tests in the case of two variables (creatinine clearance, uric acid).
Due to the statistical revisions, the corresponding sections in the Methods and Results have been re-written and the Figure 1 and Table 1 have been edited. The main conclusions of this study remained unchanged after the statistical revisions: the 12-week 5/6 nephrectomy model was characterised by increased ETA protein expression and reduced ratio of ETB:ETA protein expression in both medulla and cortex. As revealed using 2-way ANOVA, hyperuricemia had a minor increasing effect on ETB protein expression (p for OXO = 0.046, figure 1) but this effect was not reflected as changes in the ETB:ETA ratio. We have added a short discussion about this finding to the lines 278-291.
Tubulointerstitial damage index was assessed in our previous report from this 12-week nephrectomy experiment with the oxonic acid intervention (Reference 30 in the present manuscript). As suggested by the reviewer, we included this data in the Table 1, and the results and methods of the histological analyses have been added to the manuscript. The 5/6 nephrectomy model was characterised by increased tubulointerstitial damage index. It seems likely that there are viciouscircle- like links between the reduced creatinine clearance, tubulointerstitial morphological changes, and unfavorable changes in the ETB:ETA ratio in the 5/6 nephrectomy model. We have added this notion on lines 257-260.
C) As suggested by the reviewer, we now compared the Sham group only with the NX group, and the NX+intervention groups only with the NX group in the 27-week 5/6 nephrectomy study. These changes have been made to the Figure 3, Table 2 and the corresponding results section. The main conclusion of the study remained the same: the 27-week 5/6 nephrectomy model also featured with an unfavorable change in the ETB:ETA protein ratio in renal cortex, and the medullary results showed a similar trend but the findings did not reach statistical significance. As instructed, the mRNA results were removed from figure 3 to table 2 and replaced by the medullary autoradiography results in
Figure 3. The Ca/P/vit D axis does not seem to have an effect on ET-receptor expressions. Of note, our previous reports from this experiment showed that Ca/P/VitD axis had effects on vascular and renal RAS components (References 33 and 34 in the present manuscript). A short discussion on these topics was included on lines 274-278.
Tubulointerstitial damage index was included in the Table 2, and the corresponding results and discussion were added to the manuscript (lines 162-164 and 323-327, respectively). The histology methods were similar to those in the 12-week 5/6 nephrectomy study.
D) We further followed the advice of the reviewer in the statistical analyses of the adenine experiments. However, most variables in the adenine model, including the main results i.e. ET-receptor protein expressions, except for ETA protein expression, were not quite normally distributed even after log10- or lntransformation, as tested using the Kolmogorov-Smirnov test and histograms and/or the variances in the groups were not equal as tested using the Levene’s test. Thus, the assumptions of 2-way ANOVA test were not completely fulfilled. Nevertheless, to follow the approach suggested by the reviewer, we also applied 2-way ANOVA for the main results, and included the results of these analyses in the manuscript (lines 216-226). As suspected by the reviewer, 2-way ANOVA revealed minor increasing effects by especially the combination of sitaxentan+cinacalcet on ETB protein expression, as well as on ETB:ETA protein ratio. We added a short discussion about these findings to the lines 315-327. In the other analyses, we used our previous statistical methods but corrected the post-hoc comparisons, as suggested by the reviewer, i.e. Control was compared only with Ade; Ade was compared with Ade+S and Ade+C; ADE+SC was compared with ADE+S and ADE+C. Parallel corrections were made to the methods and results sections as well as Table 3 and Figure 4.
We also included a short discussion about the current findings and recent histological analyses of our previous adenine model study on lines 304-312.
We thank the reviewer for the interesting references (Lariviere et al 2017 and Ritter et al 2014) about the extrarenal positive effects of selective ETA antagonist in 5/6 nephrectomy rats. We have added these references to the discussion (lines 348-354). The overall positive findings of these and most studies of ETA antagonists in CRI are in line with our results i.e. pharmacological modulation of ETA activity can, at least partly, counteract the negative effects of the offset balance between ETB and ETA expression.
E) We have corrected the numbers of animals and laboratory samples in Table 1.
F) We have now used logarithmic transformations where appropriate, as well as Kolmogorov-Smirnov test to check the normality of every variable. However, as discussed in more detail in previous sections, some of the variables were not normally distributed even after log- or ln-transformation and/or did not have equal variances between the groups. Therefore, the use of parametric tests in these cases was not possible and we were compelled to apply nonparametric tests at some points.
G) As instructed, we corrected the first paragraph of the introduction so that ETA receptor antagonist treatment may have renoprotective effects. We replaced the previous ERA (referring also to unselective ET-receptor antagonists) by ETA receptor antagonist as pointed out by the reviewer. We agree that ETA-selective approach seems reasonable based on most previous studies and by the findings in the present study.
We included the word may in the aforementioned sentence for two reasons: First, especially in the clinical setting, there are still major safety issues with this drug group (including potential increase in edema). These aspects and a
comprehensive up-to-date review of recent literature about the renoprotective effects of ETA antagonists are found in the review article by Vaněčková et al. (Physiol Res 2018) as provided by the reviewer. We cited this comprehensive
review in the beginning of the introduction (lines 40-41). Second, the recent interesting study by Čertíková Chábová et al. (Kidney Blood Press Res 2019), also referred by the reviewer, showed that in established and extreme setting of
5/6 nephrectomy in Ren-2 renin transgenic rats (TGRs), the addition of ETA antagonist had no additional positive effects on albuminuria or survival when compared with RAS blockade alone and even abolished the positive effects of
RAS blockade plus soluble epoxide hydrolase blockade. As stated by the authors, this unexpected negative effect may indicate that the potential therapeutic value of the modulation of ETA activity may take place in the earlier phases of CRI, which may be difficult to detect in the real-life clinical settings. We have included also this reference to the manuscript (lines 270-273).